# Collision Cross Section Prediction Based on Machine Learning

**DOI:** 10.3390/molecules28104050

**Published:** 2023-05-12

**Authors:** Xiaohang Li, Hongda Wang, Meiting Jiang, Mengxiang Ding, Xiaoyan Xu, Bei Xu, Yadan Zou, Yuetong Yu, Wenzhi Yang

**Affiliations:** 1State Key Laboratory of Component-Based Chinese Medicine, Tianjin University of Traditional Chinese Medicine, 10 Poyanghu Road, Tianjin 301617, China; xiaohang0519@163.com (X.L.); 17862987156@163.com (H.W.); jiangmeiting21@163.com (M.J.); mengxiangd98@163.com (M.D.); xxy_0421@163.com (X.X.); fullmoonvn@163.com (B.X.); zouyadan01@163.com (Y.Z.); 13165576725@163.com (Y.Y.); 2Haihe Laboratory of Modern Chinese Medicine, Tianjin University of Traditional Chinese Medicine, 10 Poyanghu Road, Tianjin 301617, China

**Keywords:** ion mobility-mass spectrometry, collision cross section, machine learning, prediction, molecular descriptor

## Abstract

Ion mobility-mass spectrometry (IM-MS) is a powerful separation technique providing an additional dimension of separation to support the enhanced separation and characterization of complex components from the tissue metabolome and medicinal herbs. The integration of machine learning (ML) with IM-MS can overcome the barrier to the lack of reference standards, promoting the creation of a large number of proprietary collision cross section (CCS) databases, which help to achieve the rapid, comprehensive, and accurate characterization of the contained chemical components. In this review, advances in CCS prediction using ML in the past 2 decades are summarized. The advantages of ion mobility-mass spectrometers and the commercially available ion mobility technologies with different principles (e.g., time dispersive, confinement and selective release, and space dispersive) are introduced and compared. The general procedures involved in CCS prediction based on ML (acquisition and optimization of the independent and dependent variables, model construction and evaluation, etc.) are highlighted. In addition, quantum chemistry, molecular dynamics, and CCS theoretical calculations are also described. Finally, the applications of CCS prediction in metabolomics, natural products, foods, and the other research fields are reflected.

## 1. Introduction

Ion mobility spectroscopy (IMS), analogous to a gas-phase electrophoresis technique, enables the separation of compounds on the basis of the differences in the mobility of ions through buffer gases under the action of an electric field [1,2]. The difference in mobility is caused mainly by distinctions between the charge, shape, and size of the molecules, which leads to the differences in drift time [3,4,5]. This difference can be described by the collision cross section (CCS) value. In general, ions with the lower mass and/or more-compact structures have shorter drift times and lower CCS values; the larger the space volume and/or the higher the mass number, the greater the CCS value. This structural dependency makes the CCS value an important parameter for compound identification. The origin of IMS can be traced back to the X-ray experiments of Thomson and Rutherford in the late 19th century [6], which even predates the study of mass spectrometry (MS) by about 15 years [1]. However, because of the commercialization of ion mobility instruments, their combination had not been popularized until recently. Mobility separation occurs in milliseconds and is compatible with the modern mass spectrometers operating at microsecond scanning speeds [7]. The coupling of IMS and MS can thus provide four-dimensional structural information for component characterization, including *t_R_*, CCS, MS, and MS/MS, thereby having great potential in reducing false-positive results and improving identification confidence [8,9]. Unfortunately, the strategy of purchasing and measuring a large number of reference standards to obtain the standard CCS values is cost-prohibitive and difficult to implement in most cases. Currently, numerous CCS values can be obtained through theoretical calculation and machine-learning-based prediction without sufficient standards [10,11]. The former usually uses molecular modeling to provide the approximate structure of the molecule and then calculates CCS by simulating the interaction between the drift gas and the analyzed ions [7,12]. These methods are relatively time-consuming and require more professionalism. The latter utilizes a large data set containing the experimentally measured CCS values and structural parameters of the compounds themselves to train, validate, and test the regression models [13]. This method has the advantages of fast calculation speed and high accuracy. At present, many CCS prediction platforms are available, such as MetCCS [10], LipidCCS [14], DeepCCS [15], AllCCS [16], and CCSbase [11], etc. Previous IMS-related reviews have described either the principles of different platforms or the advantages of a specific platform. In this review, we give a comprehensive summary on both the principles and the advantages of different platforms, which can thus lay a foundation for the workflows of machine learning (ML) for CCS prediction. In addition, we focus on the general steps of constructing a CCS database on the basis of using ML algorithms (Figure 1). Notably, along with the ML model, the quantum mechanical (QM) workflows have been developed as well [17]. In this review, the commonly used methods and techniques in various links are summarized, and some practical tips are proposed.

## 2. Ion Mobility-Mass Spectrometry (IM-MS)

### 2.1. Ion Mobility Platforms with Different Separation Principles

Up to now, the commercially available mainstream IM systems are divided into three types of platforms on the basis of their separation principles: time dispersive, ion confinement with selective release systems, and space dispersive [1,4,18,19]. In current research, the first two are the most commonly used, and space-dispersive methods have higher development potential. Table 1 shows a comparison of the characteristics of different IM-MS systems.

In a time-dispersive IM-MS system, all ions drift along the same path and are detected by the detector at different times. Generally, ions with small cross-sectional areas are detected first, thanks to their high mobility. Figure 2A shows its specific working principle. The main time-dispersive techniques include drift tube IMS (DTIMS) and traveling-wave IMS (TWIMS). DTIMS consists of several ring electrodes stacked alongside that are filled with an inert static gas through which ions move as directed by a uniform electric field [20]. The drift time can be correlated directly with the CCS value through the Mason–Schamp relationship (Equations (1) and (2)) [21] without requiring a correction program [22,23]. Nevertheless, DTIMS devices have a low-resolution limitation. Researchers have in recent years taken various approaches to improve their resolution, thereby increasing the analysis range and separating isomers that have similar structures, approaches such as increasing the length of the drift tubes [24] to enhance the electric field, introducing multiplexing technology [25,26,27,28], developing a new dual drift tube IMS [29], etc. In contrast to DTIMS, ions in TWIMS are directed by a sequence of symmetric potential waves that continuously propagate through the drift region to pass through stationary gases [30,31]. The CCS values of a TWIMS instrument cannot be directly calculated on the basis of the measured drift time, because of the nonuniformly applied electric field. It needs to be calculated on the basis of a group of predefined calibrators, usually using the CCS values derived from DTIMS as a reference [23,32]. Structural similarity between calibrators and analytes is critical for achieving accurate CCS calibration [33,34]. TWIMS has a greater resolution than DTIMS with a uniform electric field for the same drift tube length [5], so TWIMS equipment take up less space while attaining a same resolution level. Recently, structures for lossless ion manipulations (SLIMs), a traveling-wave-based platform, have been developed to guide ions through a printed circuit board path, which maximizes transmission efficiency, increases path length, and achieves an extremely high resolution [35,36]. Cyclic ion mobility-mass spectrometry (cIMS) separates ions in a cyclic mobility chamber and provides significantly longer path lengths by increasing the number of times that ions pass through the cell, thereby improving the resolution and storage capacity of IM separation. The design of the cIMS device allows for IMS^n^ experiments, where ions can undergo multiple instances of selection, activation, or fragmentation and reseparation before MS detection [37]. The flexibility and practicality of the cIMS separator and control software have led to its wide application in the separation of isomers in different fields [22,38,39,40,41,42]. Time-dispersive instruments allow the simultaneous analysis of all ions and are currently widely used in untargeted metabolomics [31,43,44].
(1)$$\Omega =\frac{3ze}{{16NK}_{0}}\sqrt{\frac{2\pi }{{\mu k}_{B}T}}$$
(2)$$K_{0}=\frac{L}{t_{A}E}\frac{P}{P_{0}}\frac{T_{0}}{T}$$
where *Ω* is the rotationally averaged CCS, *K*_0_ is the reduced mobility, *z* is the charge state of the ion, *e* is the elementary charge, *N* is the number density of the drift gas, *μ* is the reduced mass of the ion–neutral drift gas pair, *k_B_* is the Boltzmann constant, *T* is the gas temperature, *t_A_* is the corrected arrival time, *E* is the electric field, *L* is the length of the drift cell, *P* is the pressure in the drift cell, and *P*_0_ and *T*_0_ are the pressure and the temperature under standard conditions, respectively.

In a confinement and selective release system, ions are driven by a parallel moving buffer gas and inversely driven by a gradient electric field. When the two forces are equivalent, the ions are stationary relative to the drift tube, indicating that they are trapped. Ions with large cross-sectional areas are stabilized in high-field regions because of their low mobility and the high electric field intensity required to maintain a static state. By reducing the intensity of the electric field, trapped ions are selectively released, and ions with a larger cross-sectional area first pass through the mobility cell and are then detected by the detector [4,45]. Figure 2B shows its specific working principle. Trapped ion mobility spectrometry (TIMS) is the most representative confinement and selective release instrument. It is no longer limited to the length of the device and can provide high resolution three to eight times larger than that of DTIMS or TWIMS [46]. Furthermore, the resolution of TIMS may be modified by adjusting custom parameters, such as the voltage scanning rate (*δ*) and neutral gas flow rate (vg), making TIMS use very selective [47]. A longer capture time can improve the resolution of the device and ion utilization. Reducing capture time, on the other hand, can enable an untargeted analysis [48]. Importantly, parallel accumulation serial fragmentation (PASEF) can be achieved by connecting two TIMS in series, one for ion accumulation and the other for ion mobility separation, which improves the duty cycles (up to nearly 100% if equal accumulation and analysis times are used in both TIMS regions) and sensitivity [49], reducing the complexity of the MS/MS spectrum [50]. Like the TWIMS, CCS values cannot be directly determined unless calibration is performed [45,51]. Thanks to its high resolution and sensitivity, TIMS, especially the PASEF strategy based on TIMS, has been applied to the separation of isomers in multiple fields [38,52,53,54].

The space-dispersive method separates ions along different drift paths on the basis of their mobility in high and low fields, but there is no significant dispersion in time. Figure 2C shows its specific working principle. Field asymmetric waveform ion mobility (FAIMS), also known as differential (ion) mobility spectrometry (DMS or DIMS), belongs to a typical space-dispersive platform [1]. The use of alternating high and low fields in FAIMS forestalls the establishment of a recognized method for obtaining its CCS values [55]. FAIMS acts as a migration filter in which only analytes that have a specific response to changing electric fields and analytes that match the applied compensation voltage can pass through the drift region and the aperture [18,56]. Therefore, FAIMS has been widely used to screen targeted metabolomics and to increase the signal-to-noise ratio of analytes of interest [13].

### 2.2. Advantages of LC-IM-MS

Recent research has shown that LC-IM-MS has advantages over conventional liquid/gas chromatography-mass spectrometry (LC/GC-MS) in the following four main aspects: (1) providing four-dimensional information to improve the characterization of isomers and enhance the reliability of identification; (2) increasing peak capacity and improving the signal-to-noise ratio (S/N); (3) obtaining additional analysis information when coupling with one or more additional analysis dimensions; (4) improving the quality of spectral acquisition [57,58].

(1)LC-IM-MS provides four-dimensional information (*t_R_*, CCS, MS, and MS/MS). As a robust parameter for characterization and recognition, CCS provides orthogonal attributes for compound recognition, improving the confidence level of compound annotation [4,59]. IMS technology has proven that it can be used to separate various isomers, such as lipid isomers [60], steroid isomers [61], fatty acid isomers [62], amino acid isomers [22], and carbohydrate isomers [63]. Numerous strategies have been introduced to enhance the IMS characterization of isomers. A combination of chemical derivatization and IMS can improve the detection of steroid isomers [61], metabolites in nicotine [64], and carbohydrates [65]. The integration of dimers or polymers with IM-MS is another effective method for identifying isomers. More accurately predicting the relative differences in CCS between steroid epimers can be achieved through the energy characteristics of the sodium dimer configuration of epimers [66]. The enantiomers of aromatic amino acids can be differentiated by TWIM-MS through their cationization with copper (II) and multimer formation with D-proline (Pro) as a chiral reference compound [67]. The mobility of ions passing through IMS is affected by using different drift gases and/or by doping volatile chiral reagents in drift gases, which can also be used to separate isomers and enantiomers [68,69]. In addition, platforms such as cIMS [42,70,71,72], multiplexed ion mobility [26,28,73], and TIMS [74] have improved the separation of isomers by improving mobility resolution. IMS can distinguish between conformational isomers [75] and isotopic isomers [22]. By taking into account all relevant errors, *N*-glycan isomers with different conformations can be distinguished on the basis of the CCS gained from the IMS [75]. As we know, lipids have a wide range of structural diversity, with a large number of isomers. A recent study used IMS to analyze the relationship between lipid structure and its gas-phase conformation, providing accurate and comprehensive conformational lipid profiles [76]. IMS has been used in the separation of isomers with different isotopic atomic positions [77] and labeled/unlabeled isotope-substituted isomers [42]. Researchers have found that IMS can be incorporated into the standard LC-MS/MS isotope analysis process as an additional separation mechanism, which can provide broader separation space and higher identification confidence for metabolic characterization [22].(2)Thanks to the advantage of increasing peak capacity and improving the signal-to-noise ratio, IMS can improve the exposure rate of trace components in complex samples [58,78]. Configuring ion mobility technology in MS studies with different ionization principles (ESI, MSI, and MALDI) can increase the peak capacity by at least two times compared with using MS alone [79,80,81]. It has been reported that when the mass resolution is 35,000 (fwhm), 860 independent ions can be measured, accounting for 15% of the total 5639 counted ions, while the addition of IMS adds 3911 features for signal recognition [79]. Because IMS is used as a separation module between LC and MS, the number of MS features detected in the metabolite composition characterization experiment has significantly increased [82]. IM-MSI can reduce chemical noise and transfer target signals from congested spectral regions, thereby increasing the S/N of metabolites and lipid peaks by nearly 10 times and doubling the image contrast [83]. Some studies have shown that compared to the traditional lipidomics methods, LC-IM-MS analysis has an increased S/N and can detect a low abundance of phospholipids in highly complex brain lipoid samples [43]. In the experiment of adding IMS to MS imaging, it was concluded that lipids with different CCS values can be spatially separated, highlighting their spatial positioning and achieving more-accurate lipid recognition [79].(3)In addition to IMS’s direct use or combining IMS with LC, it can also combine with gas chromatography (GC), mass spectrometry imaging (MSI), or supercritical fluid chromatography (SFC) technologies. As a result, multidimensional analytical information is provided, and the selection of methods increases. IMS and LC can provide orthogonal separation, with IMS separation occurring within milliseconds, and it is compatible with modern MS that is running at microsecond scanning speeds, allowing maximum separation of metabolite ions prior to MS characterization. IMS is often used in series with reverse-phase liquid chromatography (RPLC) [84,85,86] and hydrophilic interaction liquid chromatography (HILIC) [87,88,89]. Some researchers have also proposed an offline two-dimensional liquid chromatography coupled with an ion mobility-quadrupole time-of-flight mass spectrometry (2D-LC/IM-QTOF-MS) analysis strategy, achieving a comprehensive characterization of multiple components in traditional Chinese medicine [8,58,90]. In addition, a study that coupled IMS with MSI technology achieved the spatial localization of bile acids in sample tissues [91]. One study integrated ultrahigh performance supercritical fluid chromatography/quadrupole time-of-flight mass spectrometry (UHPSFC/QTOF-MS) and ion mobility spectroscopy/time-of-flight mass spectrometry (IMS/QTOF-MS) to establish a lipid omics platform for CCS measurement, which has improved the analytical performance and recognition reliability of lipids [92].(4)IM can improve the overall resolution of the spectrum and obtain high-quality MS^1^ and MS^2^ spectra. Double-charged ion clusters make the types of precursors thoroughly complex and can easily generate false positives when annotating MS^2^ data. IM is capable of separating dimers or double-charged ions in a full scan spectrum and generating high-resolution spectra of MS^1^ and MS^2^ that are close to the standards [58,84]. Wang [58] used an LC-IM-MS system to comprehensively characterize the multicomponents of compound Danshen dripping pills (CDDPs) and elucidated the advantages of IM. IM can improve the overall resolution of the spectrum of CDDPs and effectively distinguish the doubly charged saponins or the dimers of salvianolic acids, to obtain high-quality MS^1^ and MS^2^ spectra and reduce the false positives of multicomponent characterization.

## 3. Collision Cross Section Value: Dependent Variable of the Model

### 3.1. Acquisition of CCS Values

Experimental measurements [16,37,93,94,95,96,97,98,99,100,101,102] and theoretical calculations are the two main ways to obtain CCS values. The latter can adopt various strategies, including theoretical-driven methods [12,94,103,104,105,106,107,108] and data-driven methods [7,10,14,15,94,109,110,111].

The experimental CCS values are obtained by acquiring the mobility data of metabolite standards by using ion mobility platforms (DTIMS, TWIMS, TIMS, etc.) that operate under low field conditions. Because they have different principles, most of them require a dedicated calibration process to determine their CCS values [1]. Currently, the stepped-field method in DTIM, considered as the gold standard for CCS measurement, is the only method that does not require calibration to measure CCS values. Another single-field method requires the use of the linear relational equations constructed by the relationship between the CCS value and the drift time of the calibrants to calculate the CCS values [112]. In TWIMS, it is also necessary to use calibrants with known CCS values to construct a nonlinear calibration curve for both, thereby using this curve and the measured drift time to calculate CCS. The selection of calibrants should meet the following conditions: (1) ensuring good chemical stability; (2) providing wide coverage of *m/z* and CCS and uniform ion distribution; (3) forming multiple charge states; and (4) being structurally similar to the object to be analyzed [23,32,101,113,114]. At present, polyalanines and Agilent ESI-L low concentration tube mixes are widely used calibrants in DTIMS and TWIMS. Unlike the previous two calibration methods, TIMS uses known mobility (*K*_0_) calibrants to establish a linear relationship between the reciprocal of mobility (1/*K*_0_) and voltage, further obtaining the measured ion mobility and finally obtaining a CCS value after conversion. The commonly used calibrants for TIMS include perfluoro-phosphazenes [47], Agilent ESI-L low concentration tube mixes, etc. In addition, the construction of a high-precision CCS database is inseparable from the stable operation of the instrument and the calibration program. During data collection, the performance of the instrument is evaluated and its stability monitored by repeatedly measuring quality-control (QC) samples at intervals of a certain number of injections [10,14]. Some researchers have collected CCS values for a large number of metabolite standards, and CCS databases for one or more types of compounds have been constructed. Table 2 shows these specialized databases, from which we can find the following: (1) three ion mobility platforms, namely DTIMS, TWIMS, and TIMS, are involved, of which DTIMS is the most widely used; (2) various types of compounds, such as metabolites, lipids, biological samples, and drugs or drug analogs, are covered; (3) the number of CCS values obtained from this method is relatively small compared to that from the calculation and speculation method. With the deepening of research, experimental CCS databases continue to increase. However, the number of compounds in the experimental CCS database is always limited because of the limitations of the number of compound standards and ion mobility resolution.

Another method of obtaining CCS values is to use computational chemistry tools to obtain theoretical CCS values. The general process of this method is as follows: (1) obtain the three-dimensional (3D) structure and possible conformational forms of compounds; (2) use molecular mechanics, molecular dynamics, quantum chemistry (especially density functional theory, DFT), etc., to screen and optimize the conformations of compounds; and (3) select the appropriate algorithm for calculation. Avogadro [105,106,107], TINKER [121], Gaussian [105,121], NWChem [107], and SPARTAN’18 [106] are commonly used software programs that can achieve geometric optimization. The calculation of CCS values can be implemented through software such as MobCal [105,121], Collidoscope [122], IMoS [106,107], and IMPACT [106]. MobCal is the most commonly used software, and it provides three algorithms: projection approximation (PA), exact hard sphere scattering (EHSS), and the trajectory method (TM) [103]. PA is the simplest, fastest, and most widely used method that reduces scattering in 3D space to simpler, low-dimensional projections. The molecule is represented as a collection of overlapping hard spheres in PA, and the calculated CCS value is the rotational average of the projected area of this set [121]. The successive introduction of the projected superposition approximation (PSA) method [104] and the local collision probability approximation (LCPA) method [123] solved a problem: the collision between ions and gases as well as noncovalent interactions were not considered in the principle of the PA method. The EHSS method simulates the trajectory of the drift gas approaching and colliding with analyte ions [124]. The algorithm is relatively complex and is often used in the calculation of macromolecular CCS. TM is the most complex and computationally intensive method among the commonly used methods. It simulates the 3D scattering of buffer gas particles under the influence of long-range interaction potential, and it takes into account the van der Waals force and polarization interaction [103]. Based on the TM algorithm, Collidoscope uses parallel computing and trajectory parameter optimization, resulting in a significant reduction in computing time [122]. The underlying algorithm of IMoS [125] is different from that of MobCal and includes the richest CCS computing methods: PA, EHSS, DHSS, TM, and DTM. Table 2 also shows the information from the CCS database obtained through theoretical calculation methods. Zanotto [126] developed high-performance CCS computation software (HPCCS), which performs CCS calculation by using high-performance computing techniques. By using the trajectory method, HPCCS can accurately calculate CCS values for a great variety of molecules, ranging from small organic molecules to large protein complexes, using helium or nitrogen as a buffer gas with considerable gains in computer time compared with publicly available codes under the same level of theory (Table 3). CoSIMS [127] is another CCS computation-software-based multithreaded trajectory method, and it is able to calculate nearly identical CCS values as MobCal can in nearly two orders of magnitude less CPU time thanks to the various numerical methods implemented into the software, even when run on a single CPU core (Table 3). Colby [12] generated a structure and chemical property library by using molecular dynamics, quantum chemistry, and ion mobility calculations and obtained over one million CCS values by using the developed in silico chemical library engine (ISiCLE). This research reconstructed the popular MobCal code for trajectory calculation, improving the computational efficiency by more than two orders of magnitude. The method of obtaining theoretical CCS in computational chemistry has certain limitations, though: (1) a large amount of calculation and logical judgment; (2) low efficiency and a long calculation time (CCS calculation of a compound often takes several days); and (3) a large CCS error, about 3–30% [128]. Therefore, the theoretical calculation accuracy and the efficiency of CCS need to be further improved. Importantly, the accuracy of CCS calculations often depends on a variety of factors, such as different buffer gases in actual measurements and whether they are corrected, the choice of different force fields during conformation generation, and different algorithms for theoretical calculations [129,130].

CCS values can also be obtained through data-driven ML methods. Developing a CCS prediction model that is based on ML requires three components: a training data set, a prediction algorithm, and a validation data set. The training data set contains parameters representing molecular structural properties and CCS values. There are various ways to reflect molecular structural properties (commonly known as molecular descriptors), and the relevant content will be described in Section 4. The training data set can use experimental or theoretical CCS values, usually using the former. The format of the validation data set should be consistent with the training data set, but the two are independent of each other, and there is no data duplication. ML algorithms are used to construct a regression relationship between the molecular structure and the CCS values and are divided into linear and nonlinear methods, which will be described in Section 5. The general process of data modeling includes (1) the acquisition of data sets (randomly divided into a training data set and an internal validation data set, according to a certain proportion); (2) the construction of prediction models (model training, model accuracy evaluation, and model parameter optimization); (3) the validation of prediction models. Different ML models are commonly used to predict the respective CCS values of small molecule compounds and have been applied to metabolites [10,15,16], lipids [14,60], drugs [7], and food [109] and in some other fields [110,111]. It has the following advantages: (1) large prediction scale; (2) fast computing speed without consuming plenty of computing resources; and (3) a small prediction error, 1–3% [128]. Table 2 shows the information obtained from the CCS database through ML algorithms. In addition, ML prediction can be combined with computational chemistry for CCS calculation. For example, Das et al. [120] developed an efficient computational CCS workflow by using the ML model in conjunction with standard DFT methods and CCS calculations. The CCS computation protocols for the calculation of CCS were the following: the determination of the molecular state; conformation generation; conformation filtering; clustering the conformations; DFT geometry optimization; atomic charge calculations; CCS calculation; Boltzmann weighted CCS; and a predicted structure. The complete workflows could make the computation of CCS values tractable for a large number of conformationally flexible metabolites with complex molecular structures.

### 3.2. Stability Evaluation of CCS Values

As a physicochemical property of chemical compounds, the CCS value has high reproducibility and stability. 

(1)CCS values are consistent among instruments and laboratories. Numerous studies [79,95,99,117,118] have demonstrated that the measurement of CCS values for metabolites with different molecular weights on multiple TWIMS in independent laboratories (between different Vion IMSs and different SynaptG2 HDMSs, as well as between Vion IMS and SynaptG2 HDMS) is repeatable, with an RSD of CCS values within ±3%. Sarah [112] studied the reproducibility of CCS values obtained from DTIMS. Upon the completion of the analysis of 51 biologically related standards (amino acids and lipids), it was found that the interlaboratory RSD was 0.30 ± 0.16%. Some studies [23,133] have compared the CCS values measured by TWIMS and DTIMS and found that the absolute percentage error (APE) of the CCS values was within 2%.(2)CCS has stability in different substrates. Giuseppe [94] found through experimental measurements that 97% of CCS values had a mean RSD of less than 2%, which demonstrates the repeatability of CCS values in various biological matrices. To test the accuracy and precision of CCS measurements in different matrices, one study [79] compared the CCS values in the database with CCS values measured from a series of lipid extracts such as porcine brain, *E. coli*, and yeast. The results showed that CCS measurements were highly stable in different matrices.(3)CCS values have long-term robustness. One study [117] evaluated the reproducibility of the CCS values of steroid compounds after 1.5 years, and the results showed that 95.7% of the CCS values had an RSD within ±1.0%.(4)CCS also has stability at different sample concentrations. In addition, some studies have proposed some insights into how to improve the repeatability of CCS measurements, especially the high reproducibility between different ion mobility platforms [1,19,134]. For example, consistent instruments, configurations, calibration procedures, etc. are used to achieve measurement standardization; the physical theory behind ion mobility is improved so that different platforms can provide the same, physically correctly calculated CCS values without requiring calibration.

## 4. Molecular Descriptors: Independent Variable of the Model

### 4.1. Molecular Representation

Molecular descriptors (MDs) are mathematical representations of molecules calculated by a specific algorithm that converts molecular structures into numbers. MDs can be divided into (1) measured values, such as polarity, log*P*, molar refractivity, dipole moment, etc., and (2) theoretical values, which can be subdivided into constitutional, topological, geometric, electronic, and physical chemistry types [135]. In addition, there are classification methods for dividing MDs on the basis of different aspects. For example, MDs can also be divided into zero- to three-dimensional descriptors [136]. In research based on ML to predict CCS databases, MDs are often used for prediction [10,14,92,109], and molecular fingerprints (MFs) [137], and molecular quantum numbers (MQNs) [11] have also been used in some studies. MFs, which are included in MDs and are usually in the form of bit vectors, have the advantages of simple operation, fast calculation speed, and high accuracy [138]. However, because of the difficulty in variable selection, there are currently few studies applied to CCS database prediction. Yang [137] creatively used molecular fingerprints and random forest algorithms to conduct CCS prediction research and obtained a CCS database with accurate prediction capabilities (*R*^2^ = 0.95, MRE = 2.2%). The MQN system defines a simple and universal chemical space to classify organic molecules and calculate their basic characteristics, including atomic and bond types, polar groups, and topological characteristics [139]. Another study [11] found that using unsupervised clustering based on MQN to decompose chemical structure diversity can train specific and accurate prediction models for each cluster, which showed better performance than using a single model for all data training. This study has broken the limitations of the “black box” prediction model and provides interpretable results. In addition, the quantum-chemical electron ionization mass spectra (QCEIMS/QCxMS) program is the first standalone MD-based program that can predict mass spectra solely on the basis of using molecular structures as inputs [17].

### 4.2. Access to Molecular Descriptors

MDs can be obtained through specialized computing software, software that includes computing MD functionality and open-source databases or algorithms. Specialized computing software includes PaDEL-Descriptor [140,141], Dragon [142,143], alvaDesc [136,144,145], Mordred [146,147], BlueDesc [145], Chemopy [148], and ChemDesc [149]. Software Discovery Studio [150] includes the calculation of MD functions. The human metabolome database (HMDB) [10], CDK [151,152], RDkit [153], and “rcdk” package [14,60] are open-source databases or algorithms commonly used. Table 4 shows a detailed comparison of some MD acquisition approaches. Thanks to the ability to provide multiple interfaces, such as a graphical user interface (GUI) and a command line interface (CLI), and the ability to calculate plenty of MDs in parallel, PaDEL-Descriptor has become one of the best choices for open-source MD computing [146]. Dragon is another widely used software program for computing MDs. Dragon can calculate a large number of MDs and allows the calculation of disconnected structures (such as salts, complexes, etc.). Although the source code of Dragon is not open, a free and easy-to-use web version of MD (e-Dragon) computing has been developed on the basis of the older version of the software (Dragon 5.4). Further, alvaDesc software can handle full and partial connection structures, provide different unsupervised variable reduction methods, and delete descriptors with constant or missing values to reduce the number of variables, and it can conveniently divide the 33 types of provided MDs into 2D and 3D ones [109,144,145]. Mordred software can calculate a large number of MDs, and its calculation speed is twice that of PaDEL-Descriptor [146]. BlueDesc can output results in a libSVM input file format, making it easy to build SVM models. ChemoPy is a free software program to calculate 2D and 3D descriptors and can calculate 1135 descriptors. Currently, some web-based MD computing platforms have been developed, such as ChemDes and the Online Chemical Modeling Environment (OCHEM). ChemDes integrates multiple software packages such as CDK, RDKit, and BlueDesc, and it has the functions of structural optimization, molecular format conversion, and similarity calculation. OCHEM is an online version of alvaDesc [154].

### 4.3. Preprocessing and Optimization of Molecular Descriptors

The main two points that generally suitable MDs should meet are as follows: (1) the correlation between MDs should be as low as possible, and (2) they should have a good correlation with one or more properties of molecules. To accurately reflect the structure of molecules, 2D absolute configurations or optimized 3D configurations should be obtained before calculating the MDs. The 2D absolute configuration of the obtained compound can be minimized by using the MM2 method in Chem3D Ultra software to minimize the energy of the chemical structure of the molecule [150], and thus a stable molecular conformation can be obtained. After obtaining the MDs, reducing their complexity and optimizing their type and quantity, especially for compounds with similar chemical structures (such as lipids), are prerequisites for obtaining high-precision CCS value predictions. Relevant research [14] has found that through comparison, the prediction accuracy of the CCS values of optimized lipid MDs has been greatly improved (*R*^2^ = 0.9941, and *R*^2^ = 0.1322 before optimization) and the common overfitting problem in lipid prediction has also been solved. The general process of MD optimization is as follows: (1) remove the same values [60], zero values, and missing values in the data set [144]; (2) eliminate a portion of the MDs that are highly correlated [148]; and (3) gradually remove the MDs that contribute little to the regression model [14,145]. Specifically, the related MDs in the third point can be deleted by using the nearZeroVar function in the R package insert [109]. Some studies [110] have used the sensitivity analysis techniques in Alyuda NeuroIntelligence software to analyze the importance of the obtained MDs, and they ultimately obtained good CCS value prediction results (with a median relative error of less than 2%). The importance of MDs is calculated by the degree of degradation of model performance after removing the MDs. In one study [145], in extreme gradient-boosting models, the contribution of each variable to the model is calculated on the basis of the number of times that it is selected for splitting, and the square of the improvement to the model is weighted by each split. The deletion or retention of MDs is determined on the basis of their importance to the model. In order to obtain high-precision prediction results, researchers have made efforts to use a combination of 2D descriptors and new 3D descriptors [7], optimizing 3D descriptors [155], and considering the ionization states of protonated and deprotonated sites [12,145,156]. The overall trend is that the compounds used to calculate MDs are closer to the true ionization state. However, some studies [143] have found that the prediction results of 3D models are superior to 2D models in only a few cases, by comparing the impact of using 2D and 3D MDs on predicting CCS performance. Therefore, it is believed that 3D energy minimization structures are usually time-consuming, hindering the realization of high throughput [142].

## 5. Machine-Learning Algorithms

### 5.1. Different Prediction Algorithms and Prediction Platforms

Prediction algorithms are used to establish a correlation between the structure of molecules and CCS values and are divided mainly into linear and nonlinear methods (Table 5). Linear modeling methods include stepwise multiple linear analysis (SMLR), principal component regression (PCR), partial least squares (PLS) regression, and the least absolute shrinkage and selection operator (LASSO) algorithm. Common nonlinear algorithms include support vector machine (SVM), neural networks, random forest (RF), and a gradient-boosting machine (GBM).

One study [156] explored the use of MDs and chemometrics tools, namely SMLR, PCR, and PLS regression, to establish predictive models for the respective CCS values of deprotonated phenolic compounds. These methods can be used in routine metabolite identification analysis. Soper-Hopper [142] used the PLS toolbox in Matlab to conduct a PLS analysis of MD and CCS values. The results showed that by using the PLS regression model of MDs, accurate CCS values can be predicted from 2D structural information. Wang [60] developed a method based on the LASSO algorithm to predict the CCS value of lipids. In this method, a series of MDs were screened and optimized to reflect the subtle structural differences between different lipid isomers. The use of MDs and a large number of standard CCS values for lipids has significantly improved the accuracy of the LASSO model. The accuracy was externally verified by using an independent data set, with median relative errors (MREs) of <1.1%. Compared with linear regression algorithms, nonlinear modeling methods have been studied more widely. The following sections will mainly introduce the commonly used nonlinear algorithms for CCS database prediction.

SVR uses a kernel function to map the MDs of metabolites into a high-dimensional feature space, establish a hyperplane in this space, and perform high-dimensional regression between the MDs and CCS values in the training data set [10]. In order to obtain high-precision CSS value prediction results, training data sets are used to optimize the kernel function parameters of the regression hyperplane. The cost of constraints navigation (C) and gamma (*γ*) are important parameters for constant optimization. The mean absolute error (MAE), median absolute error (MDAE), median relative error (MDRE), and root mean square error (RMSE) are used as the calculation performance indicators [10,14,157]. SVR-based prediction can be achieved through the R package “e1071” (https://cran.r-project.org/web/packages/e1071, accessed on 20 February 2023) or CCSP 2.0 platform [147,158]. Zhou [10] reported for the first time a MetCCS database using the support vector regression (SVR) algorithm. This study conducted large-scale CCS predictions for 35,203 metabolites in the HMDB. Next, for the study of lipids, a stepwise elimination method was used to screen out 45 MDs that were highly correlated with CCS values. The SVR method was also used to develop a prediction CCS database containing 15,646 lipids, namely LipidCCS, with significantly improved prediction accuracy (MRE = 1%) [14]. Finally, [16] they collected more than 5000 experimental CCS values from 14 experimental data sets as a large-scale training set, and they continued to use the SVR algorithm to develop the world’s largest CCS database of different types of small molecule compounds (more than 1.6 million small molecules), which was named AllCCS.

A neural network, also known as an artificial neural network (ANN), is a type of ML. Its name and its structure are inspired by the human brain and simulate the way that biological neuron signals communicate with each other. The neural network consists of a node layer, including an input layer; one or more hidden layers; and an output layer. Each node is connected to another node and has associated weights and thresholds. If the output of any one node is higher than the specified threshold, the node will be activated and send data to the next layer of the network. The deep neural network (DNN) can be understood as a neural network with many hidden layers. A convolutional neural network (CNN) is a subtype of DNNs, consisting of a feature learning section and a prediction section [15]. It learns the internal representation of input through a series of convolution and maximum pooling steps. This internal representation is then used as an input to the multilayer perceptron to perform the prediction. CCS value prediction based on neural network algorithms can be performed on Alyuda NeuroIntelligence 2.2 software [110,133] or built using the Keras library and Tensorflow backend on the programming software Python [15,159]. Pier-Luc [15] established a neural network between the SMILES format and the CCS of compounds, successfully developed a CCS database called DeepCCS on the basis of CNNS, and predicted the CCS values of more than 2400 compounds (MDRE = 2.7%). Colby’s research team developed an algorithm, DarkChem, for metabolomics and predicting the CCS values of untargeted small molecules that is based on neural networks [159]. The algorithm used the SMILES format representing the structure of compounds as inputs and extracted CCS values and *m/z* data on compounds from the PubChem database and the ISiCLE database obtained through computational methods as output. A neural network was established to predict the various physical and chemical properties of compounds, including the CCS values. Through this training mechanism, DarkChem can predict CCS with an average error of 2.5% and can predict CCS values of nearly 600,000 small molecules.

GBM is an integrated learning method. “Boosting” refers to an iterative process that integrates multiple individual learners to form a series of weak learners into strong learners, thereby reducing model generalization errors and improving model prediction accuracy. It can be used for mathematical problems such as classification and regression [160]. At the same time, gradient boosting is mostly constructed by decision trees, also known as gradient-boosting decision trees, which have good fitting ability for linear and nonlinear data, can handle continuous and discrete data, and have high prediction accuracy and strong generalization ability. Extreme gradient boosting (XGBoost) is a scalable ML system for tree boosting, featuring efficiency and flexibility [161]. There are a few studies in which GBM algorithms is used to obtain predictive CCS databases. Nye [98] used the GBM algorithm to predict the CCS values of metabolites in their study of comparing the CCS values obtained through TWIMS and UHPLC-IMS. Connelly et al. [162] compared the experimental, theoretical, and predicted CCS values through ML for isomeric drug metabolites. The CCS value predicted by ML was obtained by using the gradient elevator algorithm, and the final prediction accuracy reached up to 2.4%. In a study by Corey [153], nearly 7325 experimental CCS values from 3775 compounds were used as dependent variables, and a prediction model for CCS values was established by using the GBM algorithm. To prevent overfitting, a nested cross-validation strategy was also used in the study. The final model value showed a mean absolute deviation of 1.2% for the data set outside the sample. Song et al. [145] compared the impact of XGBoost and the SVM algorithm on prediction models in their research on building a database of chemicals related to plastic packaging. It was found that SVM models based on CDK descriptors provided more-accurate prediction results.

Random forest (RF) is a classifier that uses multiple decision tree units to train and predict samples. It was first proposed and developed by Leo Breiman and Adele Cutler [163] and is also an integrated learning algorithm. Unlike GBM, RF uses the bagging idea, which means that the training sets of decision trees are independent of each other, and the decision trees that makeup RF can be generated in parallel with each other, which applies to both the classification and the regression problems. The RF algorithm can be implemented on the R language open-source software package randomForest (v4.6-14) [164]. The research by Ieritano found that the RF regression algorithm showed the best performance in the correlation between differential mobility and CCS values, compared to the DNN model [165]. The average absolute percentage error of the predicted CCS by RF was 2.6 ± 0.4% for analytes outside the training set. Fan Yang [137] creatively developed a cross-platform CCS value prediction method using RF algorithms and molecular fingerprints. The test accuracy of this model is above 0.85, and the median of the relative residual is around 2.2%.

**Table 5 molecules-28-04050-t005:** The algorithms for CCS prediction.

Algorithm	Method Type	Tools	Features	Refs.
Stepwise multiple linear analysis (SMLR)	linear	R package *MLRMPA*	Data need to be normalized to reduce the impact of overfitting	[156]
Principal component regression (PCR)	linear	R package *MASS*	Can reduce the dimensionality of the data set while maintaining the features with the maximum variance contribution in the data set	[156]
Partial least squares regression (PLS)	linear	Matlab with the PLS toolbox/R package *pls*	Not sensitive to multicollinearity issues caused by the use of simple linear regression models	[150,156]
Least absolute shrinkage and selection operator (LASSO)	linear	Open-source R programming	Have powerful ability to perform both variable selection and regularization	[60]
Support vector machine (SVM)	nonlinear	R package *e1071*	Wide application; relatively small sample size; can effectively avoid overfitting	[10,14,16,145,147,158]
Artificial neural network (ANN)	nonlinear	Alyuda NeuroIntelligence 2.2	Can perform supervised learning, unsupervised learning, and semisupervised learning	[15,110,133,159]
Random forest (RF)	nonlinear	The scikit-learn Python package	Low variance; low susceptibility to overfitting; poor model applicability	[137,165]
Gradient-boosting machine (GBM)	nonlinear	XGBoost library	Overfitting often occurs	[98,152,153,162]

### 5.2. Evaluation and Verification of Prediction Algorithms

The evaluation and the validation of prediction algorithms often use internal and external validation. The data set for internal validation and the training set are from the same instrument, while the external validation set uses different instruments to obtain the CCS values in the data set [7,10]. When internal and external validations are performed on the created CCS prediction model, the decision coefficient *R*^2^, mean absolute error (MAE), median absolute error (MDAE), mean squared error (MSE), and root mean square error (RMSE) are used mainly as evaluation indicators for the prediction performance of different models. Their calculation formulas are as follows:(3)$$R^{2}=1-\frac{\sum_{i=1}^{n}{\left(\mathrm{CCS}-\hat{\mathrm{CCS}}\right)}^{2}}{\sum_{i=1}^{n}{\left(\mathrm{CCS}-\bar{\mathrm{CCS}}\right)}^{2}}$$
(4)$$\mathrm{MAE}=\frac{\sum_{i=1}^{n}\left|\mathrm{CCS}-\hat{\mathrm{CCS}}\right|}{n}$$
(5)$$\mathrm{MedAE}=median\left(\left|{\mathrm{CCS}}_{1}-\hat{{\mathrm{CCS}}_{1}}\right|,\ldots ,\left|{\mathrm{CCS}}_{n}-\hat{{\mathrm{CCS}}_{n}}\right|\right)$$
(6)$$\mathrm{MSE}=\frac{\sum_{i=1}^{n}{\left(\mathrm{CCS}-\hat{\mathrm{CCS}}\right)}^{2}}{n}$$
(7)$$\mathrm{RMSE}=\sqrt{\frac{\sum_{i=1}^{n}{\left(\mathrm{CCS}-\hat{\mathrm{CCS}}\right)}^{2}}{n}}$$
where CCS represents the measured CCS value of the compound measured in LC-MS, $\hat{\mathrm{CCS}}$ represents the predicted CCS value (by ML) of the compound predicted by the constructed model, $\bar{\mathrm{CCS}}$ represents the CCS mean value of a training or verification set, and *n* refers to the number of samples in the training or verification set. The value of *R*^2^ is between 0 and 1, and the larger the value, the better. The larger the *R*^2^ of the training set, the higher the degree of fitting of the model, and the larger the *R*^2^ of the verification set, the better the prediction ability; the smaller the values of MAE, MDAE, MSE, and RMSE, the more accurate the prediction results of the model and the smaller the error.

## 6. CCS Prediction Applications

Thanks to the advantages of CCS prediction, some ML-based CCS databases have emerged one after another. These databases and self-built databases have been applied to fields such as metabolomics, natural products, food, and the other research fields (Section 6.4). Figure 3 shows the specific applications and advantages of CCS prediction.

### 6.1. In Multiomics

CCS prediction methods based on ML have been widely used in lipidomics, proteomics, and metabolomics. In 2016, Zhou et al. [10] first proposed a strategy for the large-scale calculation of metabolite CCS values using ML methods. They focused on small molecules and used the SVR algorithm to construct a regression relationship between 14 MDs and 400 measured metabolite CCS values. This study ultimately established a predictive CCS database containing 35,203 metabolites, which has high predictive accuracy. The database has also been proven to effectively improve the accuracy and efficiency of identification in untargeted metabolomics. Zhou et al. [14] then used a similar method to construct a regression relationship between the optimized molecular descriptor and more than 450 measured lipid CCS values, and they obtained a CCS value database containing more than 60,000 lipids. Notably, thanks to the high similarity among lipid structures, they used the bioinformatics methods to optimize a set of molecular descriptors and finally established a lipid CCS prediction model with high prediction accuracy. They also concluded that using the database can effectively reduce false-positive lipid identification results in untargeted lipidomics. To annotate both known and unknown metabolites in untargeted metabolomics on the basis of using IM-MS, Zhou et al. [16] developed an integrated multidimensional matching strategy. This strategy integrates over 5000 experimental CCS values and approximately 12 million CCS values predicted by ML, forming a diverse CCS database called AllCCS. The prediction method includes an optimized ML prediction algorithm, a large training data set with a high structure diversity, and a predictive performance evaluation system with representative structure similarity (RSS) score. The AllCCS database has proven to help expand the chemical coverage of identification and reveal comprehensive chemical and metabolic insights into biological processes. The DeepCCS database built by Plante and using convolutional neural network algorithms is trained and validated by using the experimental data sets of over 2400 molecules [15]. Users only need to input the SMILES symbol and the ion type of the compound to easily and quickly obtain the CCS value, which avoids the error problems that users often encounter when using MDs. DeepCCS has been proven to have high prediction accuracy, with a coefficient of determination of 0.97 and a median relative error of 2.7% over a wide molecular range. Wang [60] applied ML prediction to untargeted lipidomics, successfully predicted the CCS values of lipids, and distinguished lipid isomers, including cis–trans isomers. Specifically, a prediction method based on LASSO has been developed and used, and the molecular descriptors of lipids have also been optimized to reflect the subtle differences between their structures. Recently, Rainey [53] reported a high-precision ML algorithm (CCSP 2.0) developed on the basis of SVR models. In particular, CCSP used the open-source Mordred package to calculate a more comprehensive set of MDs. This algorithm can effectively filter false-positive results in metabolomics. Liu [9] developed a quantitative structure–retention relationship (QSRR) strategy and established a 4D information database containing *t_R_*, CCS, MS, and MS/MS for 170 important signaling lipids (*N*-acetyl ethanolamines, NAEs) by using the AllCCS database. Combining it with this database, they identified 68 NAE lipids in different biological samples.

### 6.2. In Natural Products

The method of CCS prediction can be used to characterize different chemical components in natural products. Song et al. [166] used LC-IM-HDMS techniques to characterize phenolic compounds in bearberry leaves. In this study, a strategy of comparing CCS values obtained from the literature and a database based on ML algorithms (AllCCS) with measured CCS values was added to the component identification workflow, and 88 compounds with high confidence were identified. In their study, a tolerance of 5% between CCS values measured by ML and those predicted by ML was considered acceptable. Wang [167] applied the prediction database (AllCCS) to the component characterization of Chinese traditional medicine *Cuscuta chinensis*. The CCS value predicted by ML provides more possibilities for distinguishing isomers in the absence of reference standards, with a total of 302 compounds identified or initially identified, of which 109 were not reported. With the continuous expansion of the prediction range and improvement of accuracy in the CCS database, its applications in the component characterization of natural products are becoming increasingly widespread [168,169].

### 6.3. In Foods

Using the SVM model, Song et al. [109] constructed a correlation between the MDs of 400 food contact materials and the experimentally measured CCS values. In this study, MDs and ML algorithms were optimized, and more-accurate prediction results were obtained. In the meanwhile, they evaluated the applicability of CCS values predicted by ML in the field of food packaging materials by comparing the established CCS value database for food packaging materials with three available predictive CCS values (CCSondemand, AllCCS, and CCSbase), and they found that the prediction given by CCSondemand was the most accurate. This model was eventually applied to the structural annotation of oligomers in polyamide adhesives. By combining it with the self-built prediction CCS database, the recognition confidence of 11 oligomers has been improved. Another study [118] compared the measured CCS values of mycotoxins with two ML databases (AllCCS and CCSbase). The results showed that the CCS values predicted by ML were highly correlated with the measured CCS values (Pearson r > 0.98). In the AllCCS prediction model, the prediction error for 91% of the compounds was within a percentage difference of ±5%. The CCSbase prediction model provided more-comprehensive structural coverage, resulting in lower deviations, where half of the analytes (50.3%) showed prediction errors within ±2%. The above research shows that the use of predicted CCS databases has a certain degree of credibility, which is helpful for the detection of hazardous compounds in foods. Through publicly available CCS databases, it is possible to gain a deeper understanding of the chemical components in food and its contact materials, thereby improving the effectiveness of food safety control.

### 6.4. In Other Fields

In the field of drugs and drug metabolites, research has used 2D and 3D combined MDs and established large-scale databases to train CCS prediction models on the basis of using ML, achieving high prediction accuracy [7]. Further, 3D information can predict different polymers, conformational isomers, and positional isomers. In a study on the characterization of pesticide components [110], researchers developed an accurate small molecule CCS value prediction tool that was based on ANN and empirical CCS values of 205 organic compounds. The applications of this prediction model to spinach samples have demonstrated its potential for application, which raises confidence in the preliminary identification of suspicious and untargeted pesticides. In environmental testing, Song et al.’s research [145] collected over 1000 experimental CCS values related to plastics in the literature and developed a plastic-packaging database based on SVM models. They applied this CCS database to the identification of plastic-related chemicals in rivers, reducing false-positive results and improving the recognition confidence level. In a compound identification of dust samples [170], the researchers referred to two CCS databases constructed on the basis of using ML during the identification process and evaluated the potential of predictive databases to increase the reliability of compound identification. The applications of predicted CCS values are summarized in Table 6.

## 7. Summary and Outlook

Ion mobility technology has achieved rapid growth in the past 2 decades, and commercial ion mobility platforms have emerged in an endless stream. IM-MS and its coupling with other analytical techniques have demonstrated outstanding advantages in greatly enhancing the confidence in the characterization and identification of chemical components, especially isomers, in different fields. In addition to experimental measurements and theoretical calculations, the prediction of CCS values can be more quickly and accurately achieved through ML methods, thereby establishing a dedicated multidimensional information database. Currently, some CCS databases based on different ML algorithms have been developed, such as MetCCS, LipidCCS, DeepCCS, and CCSbase. Moreover, the CCS databases have been applied in fields such as metabolomics, natural products, and foods. The development of quantum chemistry or molecular dynamics, such as the screening and optimization of 3D conformations and the determination of protonation/deprotonation sites, is helpful for obtaining the gas-phase structure closer to the measured state. The more comprehensive MD calculation methods can obtain more expected independent variables. The higher resolution of the IMS platform helps to obtain higher precision for dependent variable values. Last but not least, the newly developed appropriate feature screening approaches, ML or deep-learning algorithms, will help to greatly improve the accuracy of model fitting. With the further growth of IM-MS and the refinement of ML algorithms, it is believed that the prediction accuracy will be improved and that the database will be continuously expanded. The technology of predicting CCS values on the basis of using IM-MS and ML will also be deeply and widely used.

## Figures and Tables

**Figure 1 molecules-28-04050-f001:**
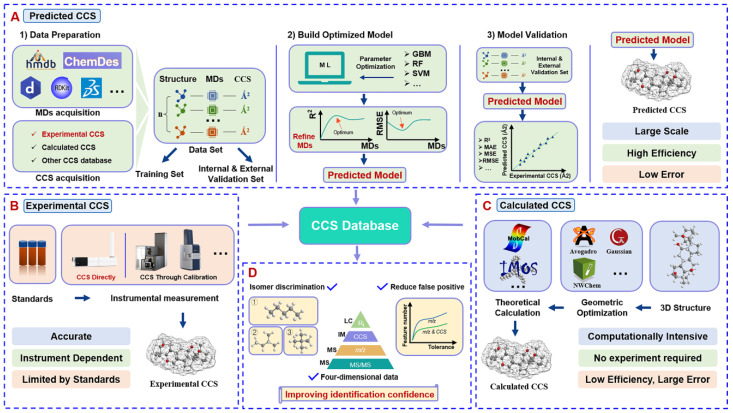
General workflow for building a CCS database: (**A**) establishing the CCS database on the basis of machine-learning-prediction methods; (**B**) elaborating the CCS database through ion mobility instrument measurement; (**C**) creating the CCS database through the theoretical calculation methods; (**D**) advantages embodied in applying the CCS database for component identification.

**Figure 2 molecules-28-04050-f002:**
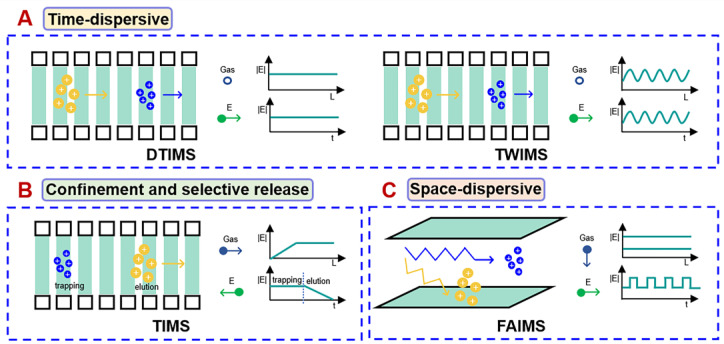
Schematic diagram of drift zone of instruments with different ion mobility values. (**A**) time dispersive; (**B**) confinement and selective release; (**C**) space dispersive. DTIMS: drift tube ion mobility; TWIMS: traveling-wave ion mobility; TIMS: trapped ion mobility; FAIMS: field asymmetric waveform ion mobility.

**Figure 3 molecules-28-04050-f003:**
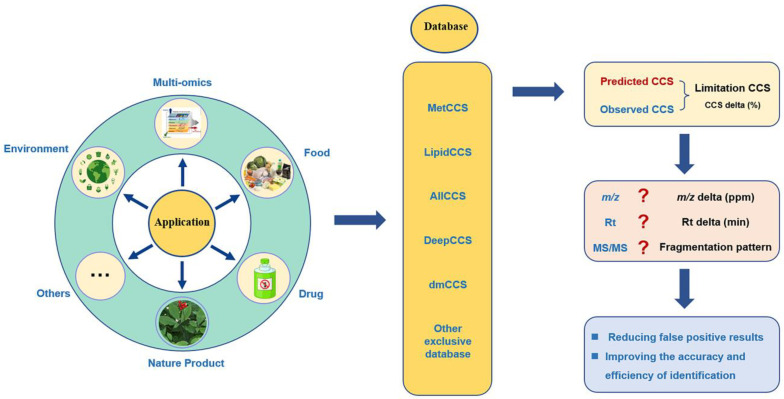
Applications and advantages of CCS prediction.

**Table 1 molecules-28-04050-t001:** Comparison of commercially available IM-MS techniques.

IMS Technique	Gas State	Resolving Power	Year of Release	CCS Calibration	Available Device	Sort
DTIMS	Stationary	~60–80	2014	Not required	Agilent IM-QTOF	Time dispersive
TWIMS	Stationary	~40–50	2006	Required	Waters Synapt HDMS Waters Vion IMS-QTOF	Time dispersive
SLIMS	Parallel gas flow	~200–300	2021	Required	MOBILion	Time dispersive
TIMS	Parallel gas flow	~200–400	2015	Required	Bruker tims TOF Bruker tims TOF pro Bruker Impact Q-TOF	Confinement and selective release
cIMS	Parallel gas flow	~750	2019	Required	Waters SELECT SERIES cyclic IMS	Confinement and selective release
FAIMS/DMS	Parallel gas flow	Not comparable	2012	-	AB Sciex SelexION	Space dispersive

**Table 2 molecules-28-04050-t002:** List of currently available CCS databases.

Source	Research Object	Number of Compounds	Number of CCS Values	Instrument Platform	Web	Ref.
Experimental CCS	Metabolites	125	209	TWIMS	/	[94]
	Lipids	244	244	TWIMS	/	[79]
	Metabolites and xenobiotics	459	826	DTIMS	http://panomics.pnnl.gov/metabolites/ (accessed on 10 February 2023)	[95]
	Primary metabolites	417	1246	DTIMS	/	[96]
	Steroids	300	1080	TWIMS	/	[97]
	Metabolites	1142	3271	DTIMS	/	[59]
	Metabolites	2193	5119	DTIMS, TWIMS	http://allccs.zhulab.cn/ (accessed on 10 February 2023)	[16]
	Metabolites	510	942	TWIMS	/	[98]
	Bile acids	47	400	DTIMS	/	[99]
	Lipids	/	594	DTIMS	/	[100]
	Lipids	1856	1856	TIMS	/	[48]
	Drug-like compounds and pesticides	~500	~500	DTIMS	/	[101]
	Small molecules	124	124	DTIMS, TWIMS	/	[23]
	Drug or drug-like molecules	1425	1440	TWIMS	/	[102]
	Doping agents	192	192	TWIMS	/	[115]
	Metabolites	112	207	TWIMS	https://massive.ucsd.edu (accessed on 10 February 2023)	[116]
	Metabolites	87	142	TWIMS	/	[117]
	Mycotoxins	53	219	TWIMS	/	[118]
	Lipids	217	456	DTIMS	https://mcleanresearchgroup.shinyapps.io/CCS-Compendium/ (accessed on 10 February 2023)	[76]
	*N*-glycans	500	500	TWIMS	/	[119]
Calculated CCS	ISiCLE: metabolites	/	~1,000,000	/	metabolomics.pnnll.gov	[12]
	Metabolites	125	205	/	/	[94]
	POMICS	/	/	/	https://www.pomics.org/ (accessed on 10 February 2023)	[120]
Predicted CCS	MetCCS: metabolites	35,203	176,015	DTIMS	http://www.metabolomics-shanghai.org/software.php (accessed on 10 February 2023)	[10]
	LipidCCS: lipids	15,646	63,434	DTIMS	http://www.metabolomics-shanghai.org/LipidCCS/ (accessed on 10 February 2023)	[14]
	AllCCS: metabolites	1,670,596	11,697,711	/	http://allccs.zhulab.cn/ (accessed on 10 February 2023)	[16]
	Pesticide residues	336	336	/	/	[110]
	DeepCCS: metabolites	2400	2400	/	/	[15]
	Sterol lipids	2068	2068	/	/	[111]
	Food contact materials	488	635	TWIMS	/	[109]
	dmCCS: drugs and their metabolites	3286	2068	/	https://CCSbase.net/dmccs_predictions (accessed on 10 February 2023)	[7]
	CCSbase: lipids, metabolites, drugs	4742	7669	DTIMS, TWIMS	https://CCSbase.net (accessed on 10 February 2023)	[11]

**Table 3 molecules-28-04050-t003:** The current CCS computation software.

Software	Year	Methods	Collision Gas	Open Source	Ref.
MobCal	1996	PA, EHSS, TM	He/N_2_	Yes	[131]
IMoS	2013	DTM, DHSS	He/N_2_	Yes	[125]
IMPACT	2015	PA	He	Yes	[132]
Collidoscope	2017	TM	He/N_2_	Yes	[122]
HPCCS	2018	TM	He/N_2_	Yes	[126]
CoSIMS	2019	TM	He	Yes	[127]

**Table 4 molecules-28-04050-t004:** Comparison between features of MD calculation software programs.

Software	Operating System	Number of Descriptors	Features	Ref.
PaDEL-Descriptor	Windows, Linux, MacOS	>1700	Supports more than 90 molecular file formats	[140]
alvaDesc	Windows, Linux, MacOS	5666	Can handle full and non-full connection structures	[144]
OCHEM	Web	5666	Is a web version of alvaDesc	[154]
chemDes	Web	3679	Integrates with multiple advanced software packages	[149]
Dragon	Windows, Linux, web (e-Dragon)	5270	Has a fast calculation speed, allowing disconnected structures	[a]
Mordred	Windows, Linux, MacOS	>1800	Can calculate macromolecule descriptor	[146]
BlueDesc	Windows, Linux, MacOS	174	Is only applicable to 3D structures	[b]
Chemopy	Windows, Linux	1135	Is applicable to 2D and 3D structures	[148]
Discovery Studio	Windows, Linux	Hundreds	Enables structural optimization	[150]
CDK	Development kit	268	Contains the chemical and bioinformatics Java library	[151]
RDkit	Development kit	200	Is based on the Python language, supporting multiple file formats	[153]
rcdk	Development kit	221	Has the CDK toolkit integrated under the R language	[106]

[a]: http://www.talete.mi.it/products/dragon_description.htm (accessed on 10 February 2023). [b]: http://www.ra.cs.uni-tuebingen.de/software/bluedesc/welcome_e.html (accessed on 10 February 2023).

**Table 6 molecules-28-04050-t006:** The applications of predicted CCS values.

Object	Year	Effect	Ref.
Metabolites	2016	MRE < 3%; the identification accuracy can be improved	[10]
	2017	MRE < 1%; the false-positive identifications of lipids can be effectively reduced	[14]
	2019	MRE < 3%; only SMILES notation and ion type are needed	[15]
	2020	MRE < 2%; the accuracy and coverage of both known metabolite and unknown metabolite annotation from biological samples can be improved	[16]
	2022	MRE < 1.1%; *cis*–*trans* and sn-positional isomers can be distinguished	[60]
	2022	MRE < 2%; the false positives can be filtered out	[147]
Natural products	2021	a higher identification confidence level can be obtained	[166]
	2022	more possibilities to distinguish isomers can be provided	[167]
Foods	2020	a certain degree of credibility can be obtained	[118]
	2022	MRE < 2%; the identification confidence of 11 oligomers can be improved	[109]
Drugs	2017	MRE < 2%; the confidence in the tentative identification of suspect and nontarget pesticides can be notably improved	[110]
	2022	MRE < 2.2%; sufficient precision to differentiate isomers and conformers can be obtained	[7]
Environment	2020	identification confidence can be increased	[170]
	2022	MRE < 2%; the false positives were reduced, and the recognition confidence levels can be improved	[145]

## Data Availability

Data sharing not applicable. No new data were created or analyzed in this study. Data sharing is not applicable to this article.

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
