# Peer review of "Collision Cross Section Prediction Based on Machine Learning"

_molecules, 2023, doi:10.3390/molecules28104050_

Round 1
Reviewer 1 Report
PDF file attached

File attached
Reviewer 2 Report
In this review, the authors summarize recent advances in machine learning-based techniques for predicting the collision cross section of ion mobility mass spectrometry. The topic is very important and meaningful and merits a systematic overview, however, the paper still has certain issues to be addressed.
Specific comments are as follows:
1. The title is prone to ambiguity. CCS can be predicted based on Machine Learning rather than Ion Mobility-Mass Spectrometry.
2. IMS is a very important technology, and a horizontal comparison and overview of its related reviews, especially papers from the last two years, need to be supplemented in the background section.
3. Because of its importance, IMS has had a number of reviews in recent years, so why the paper still has room to write is not well represented in the background.
4. The authors sorted out four advantages of IMS coupled with LC-MS, of which the fourth one is less described, whether the application is to be explored or not sufficiently described.
5. In section 3.1, the different CCS calculations (obtained) according to the instrumentation principle or computational methods can be organized in a table format with appropriate comments to facilitate the reader's quick access to the information.
6. The authors often treat ''Predicted CCS'' and Predicted CCS by ML'' as the same in the text, and the coverage of these two should be different (CCS can only be predicted by machine learning methods?), so please pay attention to the accuracy of the terminology.
7. The subheading "Molecular Descriptors" in the fourth part is not very appropriate and does not show a logical relationship with the preceding and following text (subheading).
8. Similarly, the content of the fifth section “Machine Learning Algorithms” and sixth section “CCS Prediction Application” should be sorted out in the form of two tables. The “application” should be “applications”.
9. The scientific content of Figures 1,2,3 is well drawn, but the readability and clarity (including resolution) of the images could be improved.
10. The conclusions and outlook section, especially the discussion of future trends, needs to be strengthened. Please focus on where machine learning will bring advances to the technology in the coming years, rather than generalizing about the development of IMS technology itself.
The Quality of English Language is acceptable.
